Late Jurassic theropod dinosaur bones from the Langenberg Quarry (Lower Saxony, Germany) provide evidence for several theropod lineages in the central European archipelago

Evers Serjoscha W. serjoscha.evers@googlemail.com 1
Wings Oliver 2
1 Department of Geosciences, University of Fribourg , Fribourg , Switzerland
2 Zentralmagazin Naturwissenschaftlicher Sammlungen, Martin-Luther-Universität Halle-Wittenberg , Halle (Saale) , Germany
Young Mark
Electronic publication date: 2020 Feb 6
Publication date: 2020
Volume: 8
Electronic Location ID: e8437
Received 2019 Nov 13; Accepted 2019 Dec 19
Copyright: ©2020 Evers and Wings
Copyright year: 2020
Copyright holder: Evers and Wings
License: This is an open access article distributed under the terms of the Creative Commons Attribution License, which permits unrestricted use, distribution, reproduction and adaptation in any medium and for any purpose provided that it is properly attributed. For attribution, the original author(s), title, publication source (PeerJ) and either DOI or URL of the article must be cited.
License URL: https://creativecommons.org/licenses/by/4.0/

Keywords: Theropoda, Dinosauria, Late Jurassic, Langenberg Quarry, Harz Mountains, Lower Saxony Basin

Funding: Europasaurus-Project 85 882, Volkswagen Foundation The Europasaurus-Project (grant no. 85 882) and Oliver Wings were funded by the Volkswagen Foundation within the initiative “Research in Museums”. The funders had no role in study design, data collection and analysis, decision to publish, or preparation of the manuscript.

==============================
Marine limestones and marls in the Langenberg Quarry provide unique insights into a Late Jurassic island ecosystem in central Europe. The beds yield a varied assemblage of terrestrial vertebrates including extremely rare bones of theropod from theropod dinosaurs, which we describe here for the first time. All of the theropod bones belong to relatively small individuals but represent a wide taxonomic range. The material comprises an allosauroid small pedal ungual and pedal phalanx, a ceratosaurian anterior chevron, a left fibula of a megalosauroid, and a distal caudal vertebra of a tetanuran. Additionally, a small pedal phalanx III-1 and the proximal part of a small right fibula can be assigned to indeterminate theropods. The ontogenetic stages of the material are currently unknown, although the assignment of some of the bones to juvenile individuals is plausible. The finds confirm the presence of several taxa of theropod dinosaurs in the archipelago and add to our growing understanding of theropod diversity and evolution during the Late Jurassic of Europe.

Introduction

Late Jurassic terrestrial sediments have seen a long history of fossil exploration (e.g.,  Close et al., 2018; Tennant, Chiarenza & Baron, 2018), which led to the discovery of an amazingly high number of dinosaur bearing formations (e.g., McAllister Rees et al., 2004). Despite the great dinosaur diversity known from that age (e.g.,  Lloyd et al., 2008; Barrett, McGowan & Page, 2009; Mannion et al., 2011), regional gaps in our knowledge of Late Jurassic dinosaur faunas still do exist. For example, most of Northern Germany was submerged during the Late Jurassic, resulting in an almost exclusively marine fossil record (Ziegler, 1990). A very rare exception is the Langenberg Quarry at the northern rim of the Harz Mountains where a variety of terrestrial vertebrates have been washed into the marine depositional environment from a nearby island (e.g., Sander et al., 2006; Wings & Sander, 2012; Wings, 2015). The diverse tetrapod fauna of the Langenberg Quarry is particularly famous for the occurrence of the dwarf sauropod Europasaurus holgeri, but also includes mammals, pterosaurs, turtles, crocodylians, and squamates (e.g.,  Wings & Sander, 2012; Wings, 2015). The theropod bones from the Langenberg Quarry have so far received limited attention (but see Gerke & Wings, 2016), because of the general rarity and incompleteness of theropod material. Here, we describe the exceptionally rare theropod bones from that locality for the first time. Although much of the fragmentary material can only be classified on higher taxonomic levels, the new occurrences reported herein add to our understanding of the regional tetrapod fauna and to theropod diversity in general.

Locality, geology and stratigraphy

The Langenberg Quarry near the town of Goslar, Lower Saxony, Northern Germany (Fig. 1) is a classic and well-studied locality exposing large sections of Late Jurassic shallow marine strata (Fischer, 1991; Lotze, 1968; Pape, 1970; Zuo et al., 2017). The layers consist of impure carbonates grading into marls. Tilted to a nearly vertical, slightly overturned position, the beds are quarried along strike, exposing them only in cross section and not along bedding planes. Sediment composition and invertebrate faunal content record changes in water depth and clear brackish influences, but there is no evidence of subaerial exposure (Lotze, 1968; Pape, 1970). The well dated sediments in the quarry range from late Oxfordian to late Kimmeridgian in age (Fischer, 1991; Lotze, 1968; Pape, 1970; Zuo et al., 2017). After the stratigraphic subdivision of Fischer (1991), most of the terrestrial vertebrate remains (including the sauropod dinosaur Europasaurus holgeri and all theropod bones) were found in bed 83, not in bed 93 as erroneously stated in some publications (Carballido & Sander, 2013; Marpmann et al., 2014; Sander et al., 2006). This bed is a light grey-greenish marly limestone. It has been assigned to the “Mittleres Kimmeridge”, a northwest-German equivalent to the lower part of the upper Kimmeridgian of the international chronostratigraphic time scale (Lallensack et al., 2015; Schweigert, 1999). During the Late Jurassic, the Langenberg Quarry was located in the Lower Saxony Basin that covered much of Northern Germany and that was surrounded by several paleo-islands (Ziegler, 1990), the source of the clastic components in the sediment.

Figure 1 Geographic location of the Langenberg Quarry in the Harz Mountains of Germany.

(A) Map of Germany with the Harz Mountains highlighted in grey and Langenberg Quarry (LQ) indicated by star. (B) Close-up of the Harz Mountain area with Langenberg Quarry and nearby towns indicated.

Fossil vertebrates from the Langenberg Quarry

The Langenberg Quarry is the only locality where the abundant and exquisitely three-dimensionally preserved material of the dwarfed sauropod dinosaur Europasaurus holgeri has been found (Carballido & Sander, 2013; Marpmann et al., 2014; Sander et al., 2006). The quarry also yielded a number of isolated teeth which belong to several different groups of theropod dinosaurs (Gerke & Wings, 2016) as well as natural track casts of large theropods (Lallensack et al., 2015).

Beds 83 and 73 also have produced a variety of non-dinosaurian vertebrate remains. Among them are the only known Jurassic mammals from Germany, the pinheirodontid multituberculate Teutonodon langenbergensis (Martin et al., 2016), the paulchoffatiid multituberculate Cimbriodon multituberculatus (Martin et al., 2019a), and the large morganucodontan mammaliaform Storchodon cingulatus (Martin et al., 2019b). Additionally, a three-dimensionally preserved articulated skeleton of a small pterosaur (Fastnacht, 2005), teeth and skeletons of the small non-marine atoposaurid crocodilian Knoetschkesuchus langenbergensis (Schwarz, Raddatz & Wings, 2017), various remains of marine crocodylians (Karl et al., 2006; Karl et al., 2008) and the partial skeleton of a paramacellodid lizard (Richter et al., 2013) have been reported. Diverse marine turtle material (including several skulls) comprises cf. Thalassemys sp., Plesiochelys sp., and possibly a new taxon (Jansen & Klein, 2014). Microvertebrate remains from the Langenberg yield beside many reptilian teeth (Wings, pers. obs.) a diverse fish fauna represented mainly by isolated teeth of marine chondrichthyans and osteichthyans (Mudroch, 2001; Mudroch & Thies, 1996; Thies, 1995).

Taphonomy

Almost all of the fossil material from terrestrial vertebrates (including all material described herein) was recovered after regular blasting operations in the quarry. Despite the large number of bones and teeth known from the sauropod Europasaurus holgeri, the general distribution of bones and teeth in bed 83 is rare. All of the extremely rare theropod bones were found intermingled with the mostly disarticulated E. holgeri material. All skeletal remains were accumulated in certain areas, probably lenses or channels. The bone-bearing sections of bed 83 were usually 30–50 cm thick and contained in all bone-rich areas a large number of well-rounded micritic intraclasts. The combination of bone material and intraclasts is also important for recognizing blocks of this specific layer in the quarry heap after the blasting. Because the blocks were not found in situ, it remains possible, although very unlikely, that the finds come from another bed nearby. In any case, they clearly belong to the lower part of the upper Kimmeridgian.

Materials and Methods

The present work is based on several isolated bones, which have been morphologically examined by the authors. Comparisons have been made on the basis of first hand observation on relevant material by one of us (SWE), as well as literature comparisons.

Results

In the following section, we describe each specimen, provide its systematic identification, and justify the latter in a remarks section by comparative notes.

Dinosauria Owen, 1842	
Theropoda Marsh, 1881	
Tetanurae Gauthier, 1986	
Avetheropoda Paul, 1988	
Allosauroidea Marsh, 1878; Currie & Zhao, 1993	

Material: DfMMh/FV1/19, small pedal ungual (Figs. 2A–2D).

Figure 2 Isolated theropodan phalangeal elements from the Langenberg Quarry.

DfMMh/FV1/19, pedal ungual, in (A) dorsal view, (B) ventral view, (C) left lateral view, (D) right lateral view. DfMMh/FV/343, pedal phalanx, in (E) dorsal view, (F) ventral view, (G) distal view, (H) left lateral view, (I) right lateral view, (J) distal view. DfMMh/FV2/19, pedal phalanx, in (K) dorsal view, (L) ventral view, (M) distal view, (N) left lateral view, (O) right lateral view, (P) distal view. Abbreviations: cg, collateral groove; ext, extensor tubercle; fxf, flexor fossa; fxt, flexor tubercle; lp, ligament pit. All scale bars equal five mm.

Description: DfMMh/FV1/19 is an ungual that measures 23 mm in a straight line from the extensor tubercle to the distal tip. DfMMh/FV1/19 is relatively slender, and ventrally only moderately broader than dorsally. The ungual has a transversely expanded proximal surface for the articulation with the preceding phalanx, and a moderately recurved body that extends distally into a sharp tip.

The proximal surface of DfMMh/FV1/19 is vaguely triangular in shape, transversely narrow, and dorsoventrally taller than wide. Its maximal height is eight mm, and the maximal width is six mm. The proximal surface is slightly damaged at the ventral rim, but the overall shape is discernible as only the surface of the element seems to be superficially broken. The margin around the proximal surface is developed as a salient rim ventrally to the extensor tubercle. While the surface of the ungual is generally smooth, the surface around this proximal rim is roughened. The extensor tubercle forms a short proximally overhanging tip at the dorsal margin of the proximal surface, and bears weak longitudinal striations on its dorsal surface (Figs. 2A–2D). The latero- and medioventral edges of the proximal surface form protruding flanges, expanding the ventral part of the proximal surface transversely in relation to the dorsal margin (Fig. 2B). The articulation facet for the preceding phalanx on the proximal surface is dorsoventrally only weakly concave and lacks a distinct vertical median ridge, although the central portion of the facet is slightly raised in comparison to the parts of the facet near the outer margins. The dorsolateral and dorsomedial portions of the proximal facet are gently deepened, indicating that the distal surface of the preceding phalanx was slightly ginglymoid.

The body of the ungual is ventrally curved, and tapers to a sharp distal tip (Figs. 2C–2D). The dorsal surface of the body of the ungual is continuous with the surface of the extensor tubercle. This surface is transversely strongly convex and smooth. On the lateral and medial side, the body of the ungual is separated from the proximal surface by a low depression, which gives the claw a slightly constricted morphology just distal to the proximal articulation.

Distal to this constriction, the ventral surface of the body of the ungual is weakly broader than the dorsal surface. The ventral surface is also slightly less ventrally curved than the dorsal margin, and is transversely almost flat for most of its length, and just slightly transversely convex near the tip pf the ungual. In the proximal part, distal to the proximal facet and separated from its margin by a shallow transverse groove, the ventral surface of the claw exhibits a small mount-like structure, the flexor tubercle (Figs. 2B–2D). As parts of the ventral surface of the claw are damaged toward the proximal articulation, the distal and left side of the flexor tubercle cannot be described. On the right side, however, there seems to be a small oblique groove or elongate depression that separates the flexor tubercle from the margin of the proximal facet.

The lateral and medial surface of the ungual are each incised by a deep groove (Figs. 2C–2D), which separates the body of the ungual into a broadened ventral part and a dorsal part. The collateral grooves parallel the ventral margin of the claw, and are therefore ventrally concavely rounded. At its proximal end, each groove merges with the medial and lateral depression, respectively. Each collateral groove starts proximally on a central position on the lateral and medial surface, respectively, but continues distally to a slightly dorsoventrally higher position, so that the broad ventral part of the claw is relatively prominent distally.

Remarks: Precise identification of DfMMh/FV1/19 is difficult, as unguals are generally not described in detail in the literature. We tentatively identify DfMMh/FV1/19 as belonging to a theropod dinosaur. Unfortunately, unguals of alternative taxa, such as crocodiles, lizards, and testudinids, all groups for which fossils have been found and described from the Langenberg Quarry (Thies, Windorf & Mudroch, 1997; Karl et al., 2006; Karl et al., 2008; Jansen & Klein, 2014; Richter et al., 2013; Schwarz, Raddatz & Wings, 2017), are even less often described in the literature than theropod unguals, so that the following comments are largely based on personal observations.

Testudinid taxa that appear in the Lower Saxony Basin, including the Langenberg Quarry (e.g., Jansen & Klein, 2014), and that are common more generally in coastal and shallow marine settings in the Late Jurassic belong to an enigmatic array of eucryptodiran taxa known as eurysternids, plesiochelyids, and thalassemydids (Anquetin, Püntener & Joyce, 2017; Evers & Benson, 2019). Eurysternids, a eucryptodiran group of Late Jurassic, secondarily marine turtles are known from many relatively complete specimens that often include manual and pedal unguals. Eurysternids generally have manual and pedal unguals that are more robust, i.e., anteroposteriorly short but transversely broad (e.g., Eurysternum wagleri BSPG AS I 921, BSPG 1600 VIII 43, SMNS 59731; Solnhofia parsoni JM SCHA 70; Joyce, 2000; Anquetin & Joyce, 2014).

DfMMh/FV1/19 exhibits some features that are comparable to claws of theropod dinosaurs. Manual and pedal unguals in theropod dinosaurs generally vary relatively strongly in morphology. Pedal unguals usually exhibit a weaker degree of curvature, are transversely broader and ventrally flatter than their manual counterparts. They also have less strongly developed extensor and flexor tubercles, and often lack a distinct median vertical ridge on the proximal articulation facet (e.g., Allosaurus fragilis: Madsen, 1976; Eustreptospondylus oxoniensis: Sadleir, Barrett & Powell, 2008; Australovenator wintonensis: Hocknull et al., 2009). In manual unguals, the median ridge is generally well developed, and separates a medial and lateral surface for the respective cotyles on the strongly ginglymoid distal articulation surfaces of penultimate manual phalanges (e.g., Australovenator wintonensis: (White et al., 2012). These surfaces are dorsoventrally usually quite tall, and the entire proximal surface is laterally and medially not much expanded in respect to the distal part of the claw so that manual unguals appear laterally compressed. Additionally, the flexor tubercle is pronounced in manual unguals (Rauhut, 2003a). The features described above for DfMMh/FV1/19 are congruent with the generalised features of theropod pedal unguals, and thus we interpret DfMMh/FV1/19 to represent such an element. However, it remains unclear if DfMMh/FV1/19 represents a right or left element, and we are also uncertain about the digit identity of DfMMh/FV1/19.

Isolated teeth of Late Jurassic theropod dinosaurs from the Lower Saxony Basin in Northern Germany, including material from the Langenberg Quarry, have been identified by multivariate and cladistics analyses as belonging to basal Tyrannosauroidea, Allosauroidea, Megalosauroidea, and Ceratosauria (Gerke & Wings, 2016). These taxa therefore provide potential comparative clues about the taxonomic identification of DfMMh/FV1/19.

Pedal unguals of non-abelisaurid ceratosaurs, such as Limusaurus inextribacilis (IVPP P 15923), are more robust, less recurved, and dorsoventrally more flattened as well as transversely broader than seen in DfMMh/FV1/19. DfMMh/FV1/19 also does not compare well with abelisaurid ceratosaurs. Pedal unguals reported for abelisaurids commonly show a broad triangular depression on the ventral surface (e.g., Majungasaurus crenatissimus: Carrano, 2007; Eoabelisaurus mefi: Pol & Rauhut, 2012), and at least some forms have a pair of collateral grooves on either side of the claw body (e.g., Masiakasaurus knopfleri: Carrano, Sampson & Forster, 2002). All of these features are absent in DfMMh/FV1/19.

Pedal unguals of basal tyrannosauroids also differ substantially from DfMMh/FV1/19. The proceratosaurid tyrannosauroid Guanlong wucaii (IVPP 14532) has relatively large extensor tubercles, collateral grooves that deepen dorsoventrally toward the proximal end of the ungual, and proximal articulations that are much more strongly concave than seen in DfMMh/FV1/19. In the basal tyrannosauroid Dilong paradoxus (IVPP V 11579), the pedal unguals are proximally dorsoventrally much deeper than in DfMMh/FV1/19, right and left sub-facets are separated by a moderately strong median ridge, and flexor tubercles are much more prominent, significantly expanding the depth of the proximal part of the unguals ventrally.

In megalosauroid theropods, pedal unguals are general more robust that seen in DfMMh/FV1/19. For instance, Eustreptospondylus oxoniensis has a preserved pedal ungual that is transversely broader in regard to DfMMh/FV1/19 both at the dorsal and ventral margins of the ungual body (OUMNH.J 13558: Sadleir, Barrett & Powell, 2008). Additionally, the extensor tubercle is more prominent and the proximal articulations facet is oval rather than triangular.

DfMMh/FV1/19 has superficial similarities to Allosaurus fragilis, in that the degree of curvature is similar, weak flexor tubercles and relatively flat ventral surfaces are present, and right and left sub-facets on the proximal articulation are only weakly differentiated (e.g., UMNH VP 5355, 5365, 5368, 6771). However, in Allosaurus fragilis pedal unguals are generally dorsoventrally higher than in DfMMh/FV1/19, have slightly more expanded extensor tubercles, somewhat more laterally compressed proximal articulation facets, and the collateral grooves are positioned more dorsally on the claw body.

DfMMh/FV1/19 is closest in both overall similarity as well as detailed aspects of morphology to material described for the neovenatorid Australovenator wintonensis (Hocknull et al., 2009; White et al., 2012). It should be noted that the unguals of Australovenator wintonensis are likely the best described unguals for any theropod dinosaur, as all ungual elements are described separately, figured, and 3D models that were created on the basis of CT scans were made available (White et al., 2012). The pedal unguals of the fourth digit (IV-5) of Australovenator wintonensis but also Neovenator salerii (Brusatte & Benson, 2013) are very similar to DfMMh/FV1/19: the respective specimens share a similar degree of curvature; a relatively flat ventral surface; a short extensor tubercle and rim around the proximal articulation facet; a gently concave facet with incomplete separation of medial and lateral sub-facets by a shallow central tubercle; a mediolaterally slightly constricted area between the proximal surface and the body of the claw; and shallow a depression to either side of the flexor tubercle. However, there are also important differences to neovenatorid theropods: Australovenator wintonensis and Neovenator salerii have a more prominent extensor tubercle than DfMMh/FV1/19 and a ventral part of the claw body that is transversely broader in relation to the dorsal portion of the claw. Additionally, the rim around the articular facet is much narrower in Australovenator wintonensis and Neovenator salerii. DfMMh/FV1/19 is also similar to the pedal ungual of the second digit (II-3) of Australovenator wintonensis. In this element, the ventral surface of the claw is less expanded transversely than in IV-5, which is more like the morphology of DfMMh/FV1/19. However, in II-3, the collateral grooves on the claw body are less deep than in both the IV-5 of Australovenator wintonensis or DfMMh/FV1/19. The pedal ungual of the first digit (I-2) of Australovenator does not match the morphology of DfMMh/FV1/19 well, as this element has a more prominent extensor tubercle, and a dorsoventrally high ovoid proximal surface as well as only faintly developed collateral grooves. The ungual of the third digit (III-4) of Australovenator wintonensis seems proximally distorted, so that the degree of similarity to DfMMh/FV1/19 is harder to establish.

Based on these anatomical observations, a tentative identification of DfMMh/FV1/19 as a pedal ungual of a neovenatorid theropod seems plausible. However, several arguments cast doubt on this interpretation. For instance, DfMMh/FV1/19 is relatively small in general terms of neovenatorids, which commonly achieve body masses of more than one metric ton and femoral lengths of around 750 mm (e.g., Neovenator salerii; Benson et al., 2014), so that it is highly likely that DfMMh/FV1/19 represents a juvenile, if it were a neovenatorid. Secondly, our interpretation could be biased by the availablility of extremely detailed ungual descriptions for neovenatorid theropods. Additionally, a neoventorid interpretation would result in a substantial range extension for the clade, as neovenatorids are thought to have evolved during the Early Cretaceous (Benson, Carrano & Brusatte, 2010). Basing a range extension into the Late Jurassic on our fragmentary remains seems problematic. Further and more complete material is needed to test the possible presence of neovenatorid theropods in Northern Germany. Here, we stick with a more conservative approach and simply refer the specimen to Allosauroidea, as we are reasonably confident with this assignment given the fragmentary nature of the material.

Dinosauria Owen, 1842	
Theropoda Marsh, 1881	
Tetanurae Gauthier, 1986	
Avetheropoda Paul, 1988	
cf. Allosauroidea Marsh, 1878; Currie & Zhao, 1993	

Material: DfMMh/FV/343, small pedal phalanx (Figs. 2E–2J).

Description: DfMMh/FV/343 is a small proximal pedal phalanx that measures 11 mm in length. The proximal articular facet of the phalanx is seven mm wide, narrows from ventral to dorsal and is three mm tall. Overall, the element is stout in dorsoventral view due to its relatively large width, but appears much more slender in lateral view due to its relatively short dorsoventral height. A small extensor turbercle is developed at the dorsal surface of the proximal end (Figs. 2E, 2H). The phalangeal shaft of DfMMh/FV/343 is somewhat more slender than the articular ends of the element, and the ventral surface of the shaft is flattened. Proximally, a broad flexor fossa is well-developed (Fig. 2F). The anterior end of the element is developed as a six mm wide trochlea that is not subdivided into distinct left and right cotylar facets (Fig. 2J). Ligament pits are developed on each of the lateral cotyle surfaces. These pits are relatively deep on both sides in DfMMh/FV/343, and do not occupy the entire lateral surface of the cotyles. On the dorsal surface, a distally placed extensor groove (for the following phalanx) is absent.

Remarks: Similarly to the pedal ungual, interpretation of DfMMh/FV/343 is complicated by the fact that the phalangeal morphology of many taxa occurring in the Langenberg Quarry is not well described. However, the general morphology of DfMMh/FV/343 is consistent with that of a pedal phalanx of a theropod dinosaur. The relatively broad overall shape of DfMMh/FV/343, as well as the presence of a singular proximal articulation facet that is not divided by a vertical ridge in subfacets or the presence of only moderately development extensor tubercles is generally typical for proximal pedal phalanges of theropods (e.g., Madsen, 1976). Although little has been explicitly published on pedal phalanx morphology for theropods, a few comparisons can be made, and which indicate potential allosauroid affinities of the material. For instance, the collateral ligament pits of DfMMh/FV/343 are deep but relatively small, as for instance in Australovenator wintonensis (White et al., 2012) and Allosaurus fragilis (Madsen, 1976), whereas the pits occupy the entire cotylar surface and are smore shallowly sloping, funnel like depressions in Eustreptospondylus oxoniensis (Sadleir, Barrett & Powell, 2008). The absent distal extensor grooves of DfMMh/FV/343 are unusual for allosauroids and theropods more widely, but it should be noted that these grooves are usually weakest in proximal phalanges (e.g., White et al., 2012)

DfMMh/FV/343 was found in the same block of matrix as the pedal phalanx DfMMh/FV1/19, although not in articulation or particularly close association with it (pers. comm., Nils Knötschke). Both specimens have are relatively small, with the ungual being somewhat longer than the non-ungual phalanx. In neovenatorid theropods, unguals are equally long or longer than more proximally positioned phalanges in the fourth digit (e.g., Brusatte, Benson & Hutt, 2008), which is consistent with one of the possibly identifications of the ungual. Therefore, it is possible (but highly speculative) that both elements belong to the same individual. Although we find some indications that the pedal phalanx could belong to a neovenatorid, we more conservatively assign it to Allosauroidea.

Dinosauria Owen, 1842	
Theropoda Marsh, 1881	
cf. Ceratosauria Marsh, 1884	

Material: DfMMh/FV/776, anterior chevron (Figs. 3A–3E).

Figure 3 Isolated Theropoda axial elements from the Langenberg Quarry.

DfMMh/FV/776, chevron, in (A) anterior view, (B) left lateral view, (C) posterior view, (D) right lateral view, (E) anterodorsal view on proximal articular surface. DfMMh/FV/105, distal caudal vertebra, in (F), ventral view, (G) dorsal view, (H) anterior view, (I) posterior view, (J) left lateral view, (K) right lateral view. Abbreviations: ak, anterior keel; hc, haemal canal; poz, postzygapophysis. Scale bar in A–E equals 20 mm, scale bar in F–K equals 10 mm.

Description: DfMMh/FV/776 is a chevron from an anterior position within the caudal vertebral series. The chevron consists of a well-preserved haemal arch and an incompletely preserved haemal spine. The haemal spine is crushed in the distal third of its preserved length, and the distal tip is broken and not preserved. The preserved parts of DfMMh/FV/776 measure c. 110 mm.

The haemal arch consists of two lateral processes that border the haemal canal, and a proximal articular surface that is buttressed by the lateral processes. The lateral chevron processes and the haemal spine are angled strongly posteriorly in respect to the articulation surface (Figs. 3B, 3D). The haemal canal is vaguely triangular and proximally broader than distally (Figs. 3A, 3D). It measures 12 mm across its widest part, and is 16 mm high. The lateral processes are anteriorly expanded to convex flanges that expand the lateral wall of the haemal canal anteriorly. These flanges are relatively small, and form symmetrically rounded anterior margins (Figs. 3B, 3D). A similar, yet much less prominent posterior expansion of the lateral processes is present. The articulation surface of DfMMh/FV/776 is anteroposteriorly narrow, measuring 11 mm, and transversely broad, measuring 35 mm. The articulation facet is not subdivided into anterior and posterior subfacets for the preceding and successive caudal vertebral articulations, but the topology of the surface is also not uniform (Fig. 3E). Instead, the articulation surface is convexly rounded to either side laterally and concavely depressed centrally. The margin surrounding the articulation surface is gently elevated, which is particularly prominent on the posterior side.

The haemal spine is elongate and slender, and has approximately parallel anterior and posterior margins. It forms a straight process that is not posteriorly kinked or curved. The transverse (=mediolateral) width of the haemal spine decreases from eight mm proximally, to four mm at its broken distal end. DfMMh/FV/776 has a low but prominent median keel on the anterior surface of the haemal spine (Fig. 3A). The keel has a sharp margin and is deepest proximally, where it forms a low anteriorly projecting flange. The proximal part of the posterior surface of the haemal spine shows a broad groove, which is continuous with the posterior opening of the haemal canal. This groove gets shallower distally and is replaced by a low median keel in the central parts of the haemal spine. The posterior keel becomes more prominent distally, and develops to a ridge-like posterior margin in the distal third of the preserved haemal spine length.

Remarks: DfMMh/FV/776 can be identified as an anterior chevron of a large theropod dinosaur, because of the presence of anterior flanges of the lateral process, which are only present in theropods (Rauhut, 2003a). Because of its rod-like haemal spine, DfMMh/FV/766 is a chevron from the anterior part of the caudal axial series. Theropod chevrons are not well described in the literature, making a precise taxonomic assessment of DfMMh/FV/776 difficult. However, a few general comparisons can be made. The relatively robust, weakly posteriorly oriented haemal arch is widespread among neotheropods, including non-tetanurans (e.g., Dilophosaurus wetherilli: Welles, 1984; Ceratosaurus sp.: Madsen & Welles, 2000) and tetanurans (e.g., Allosaurus fragilis: Madsen, 1976). Most theropods show a subdivision of the articular facet into an anterior and a posterior subfacet, which are usually separated by a transverse ridge. DfMMh/FV/766 lacks such a subdivision, and instead has a single articular facet. Carrano, Benson & Sampson (2012) found chevrons without a subdivided facet but low lateral mounds on each side as a putative synapomorphy of Megalosauroidea (e.g., present in Torvosaurus tanneris, Baryonyx walkeri, Afrovenator abakensis). However, undivided articulation facets have also been described for chevrons of ceratosaurs (Bonaparte, Novas & Coria, 1990; Coria & Salgardo, 1998; O’Connor, 2007), some of which show this feature only in the first chevron (e.g., Majungasaurus crenatissimus: (O’Connor, 2007). This indicates that the character has a wider distribution than recognized by Carrano, Benson & Sampson (2012). The relatively small size of the anterior flanges of DfMMh/FV/776 matches the condition described for ceratosaurs better than for megalosauroids, in which the flanges are either absent altogether (e.g., Baryonyx walkeri: Charig & Milner, 1997), or relatively pointed (e.g., Torvosaurus tanneri: (Britt, 1991). Allosauroids usually have more prominent and anteriorly pointed anterior flanges (e.g., Madsen, 1976; Zanno & Makovicky, 2013; Malafaia et al., 2016), making it unlikely that DfMMh/FV/776 represents an allosauroid. The straight haemal arch of DfMMh/FV/776 is also compatible with ceratosaurian affinities (e.g., Ceratosaurus sp.: Madsen & Welles, 2000; Carnotaurus sastrai: Bonaparte, Novas & Coria, 1990), but is also observed in some megalosaurs such as Torvosaurus tanneri Britt, 1991). The anterior median keel observed ofr DfMMh/FV/776 is peculiar, as it is a relatively prominent feature. We have not seen a comparative ridge in allosauroids (e.g., Madsen, 1976), and the keel also seems to be absent in megalosaurs such as Torvosaurus tanneri (Britt, 1991). Carnotaurus sastrai seems to have a weak anterior ridge (Bonaparte, Novas & Coria, 1990), but this feature is much more pronounced in. Based on these limited comparisons, DfMMh/FV/776 is most compatible with the chevron morphology observed in ceratosaurs, although megalosauroid affinities cannot be ruled out entirely.

Dinosauria Owen, 1842	
Theropoda Marsh, 1881	
Tetanurae Gauthier, 1986	
cf. Megalosauroidea Fitzinger, 1843; Walker, 1964	

Material: DfMMh/FV/287, left fibula (Figs. 4A–4E).

Figure 4 Isolated theropodan fibulae from the Langenberg Quarry.

DfMMh/FV/287, left fibula in (A) anterior view, (B) medial view, (C) proximal view, (D) posterior view, (E) medial view. DfMMh/FV3/19, partial right fibula in (F) anterior view, (G) lateral view, (H) proximal view, (I) posterior view, (J) medial view. Abbreviations: g, groove; mf, medial fossa; pr, posterior ridge; tif, tubercle for the M. iliofibularis. All scale bars equal 20 mm.

Description: DfMMh/FV/287 is a partially preserved left fibula, in which the distal end is missing. The proximal part of the fibula is expanded anteroposteriorly to form the fibular head, and distally the element forms a slender shaft (Figs. 4A–4E). The expansion of the fibular head relative to the shaft is asymmetric, and proportionally stringer on the posterior side: The posterior margin of the fibular head forms a convexly rounded process, which is separated from more distal parts of the fibula by a gentle notch. The posterior margin of the fibular head is formed as a transversely thin edge. In contrast, the anterior side of the fibular head expands more gradually, and forms a relatively thick and rounded margin. The fibular head is inflected medially (i.e., toward the tibia) at its anterior side, giving the proximal articular surface a crescentic outline (Fig. 4C). On the medial surface, the fibular head bears a shallow, concave fossa (Fig. 4E). The fossa is limited to the anterior and central parts of the medial surface, and does not extend onto the thin posterior expansion of the fibular head. Anteriorly, the fossa is well defined by a vertically projecting anteromedial ridge. The fossa extends distally to the level of the tubercle for the M. iliofibularis, and is developed as a deep trough just proximally to the tubercle. It remains unclear, if the present depth of the fossa is natural in this part of the bone, of if slight crushing hypertrophied this structure.

The proximal part of the fibula tapers distally to the level of the notch, until it reaches the tubercle for the M. iliofibularis. This tubercle is developed as a bulbous swelling with a rugose surface texture, which makes the fibula appear expanded in this part (Fig. 4E). The tubercle is located on the anteromedial side of the bone, but covers most of the medial surface of the fibula as well.

Distally to the tubercle for the M. iliofibularis, the fibular shaft extends as a slender and rod-like structure. The fibular shaft is kinked posteriorly in respect to the proximal third of the fibula, best seen in lateral or medial view, but is itself straight. The shaft retains its width and depth along the rest of its preserved length. The fibular shaft is slightly longer anteroposteriorly than it is transversely wide, and it is conspicuously concavo-concex, whereby the lateral surface of the cone is strongly convexely rounded. The medial surface of the shaft is furrowed by a longitudinal groove. The distal end of the fibula is broken off, so that it is unclear if and how the bone expands to articulate with the tarsus.

Remarks: The fibula DfMMh/FV/287 can be assigned to the Theropoda because of the presence of a marked M. iliofibularis tubercle, which is only found in theropods among dinosaurs (Rauhut, 2003a; Rauhut, 2003b). Within Theropoda, DfMMh/FV/287 represents a member of the Averostra, because of the presence of the fossa on the medial surface of the fibular head, which is absent in earlier branching lineages such as coelophysids (Rauhut, 2003a). It is unlikely that DfMMh/FV/287 belongs to a ceratosaur, because the fossa on the medial surface in these theropods is usually very deep and anteriorly and posteriorly bound by well-defined ridges (e.g., Ceratosaurus sp.: UMNH VP 5278, Madsen & Welles, 2000); Eoabelisaurus mefi: (Pol & Rauhut, 2012); Elaphrosaurus bambergi: (Rauhut & Carrano, 2016). Additionally, the medial surface of the shaft is not concave in ceratosaurs such as Ceratosaurus (UMNH VP 5278), whereas this is the case in basal tetanurans, including allosauroids (e.g., Allosaurus fragilis: UMNH VP 7949) or megalosauroids (e.g., Afrovenator abakensis: UC OBA 1). Among the Tetanurae, DfMMh/FV/287 is most similar to megalosauroids, some of which share the unusually shallow medial fossae with DfMMh/FV/287 (e.g., Suchomimus tenerensis: MNN GDF501; Torvosaurus tanneri: BYU VP 9620; (Britt, 1991; Benson, 2010). In allosauroids, such as Allosaurus fragilis (Madsen, 1976), Neovenator salerii (MIWG 6348; Brusatte, Benson & Hutt, 2008), or Australovenator wintonensis (White et al., 2013), the medial fossae are usually deeper, and proximally well bound by a sharp margin, whereas the fossa on DfMMh/FV/287 simply becomes shallower proximally.

Dinosauria Owen, 1842	
Theropoda Marsh, 1881	
cf. Tetanurae Gauthier, 1986	

Material: DfMMh/FV/105, distal caudal vertebra (Figs. 3F–3K).

Description: DfMMh/FV/105 is identified as a distal caudal vertebra based on its general centrum dimensions and development of neural arch processes. The vertebra is incomplete, as the posterior intervertebral articulation is ventrally splintered, obscuring most of the ventral surface, and the prezygapophyses are broken at their bases. However, the delicate postzygapophyses are preserved.

The centrum of DfMMh/FV/105 is anteroposteriorly elongate and measures 33 mm in length. The intercentral articulations are thus broader than high (12 mm wide and nine mm high for the anterior facet; Fig. 3H), and reniform in shape. The facets are slightly concave, and the vertebra is accordingly amphiplatian/amphicoelous. The exposed anterior intercentral articulation is dorsoventrally and transversely expanded in relation to the mid-centrum, so that the centrum is centrally gently constricted. The lateral surface of the centrum bears a longitudinal ridge on either side, which is sometimes observed in posterior caudal vertebrae past the transition point, i.e., the vertebral position after which the transverse processes are fully reduced in theropod dinosaurs. A cross-section through the central part of the centrum would be hexagonal because of the lateral ridge on either side.

The neural arch is low and elongate (Figs. 3F–3K), but does not cover the centrum from end to end. Instead, the neural arch is removed from the dorsal margin of the anterior and posterior intercentral articulations, exposing a broad floor of bone anterior to the entry and posterior to the exit of the neural canal. The prezygapophyses are broken off, but their remaining pedicles suggest that they were larger than the postzygapophyses, and diverged slightly from the midline, as commonly found in theropods. The neural arch between the pre- and postzygapophyses is a continuously low table of bone, which is transversely narrower than the centrum.

The postzygapophyses are delicate processes (Figs. 3J–3K), which are only weakly diverged from the midline. They overhang the posterior end of the centrum by a few millimeters. The postzygapophyses are slightly twisted, so that their articulation facets point progressively more ventrally as they are approaching their distal tip; while the articulation facets are basically laterally oriented at the base of the postzygapophyses, the facets have a strongly lateroventral inclination at their tips. This suggests, that the (unpreserved) prezygapophyses reached far onto preceding vertebrae, as is the case for many theropods.

The dorsal surface of the neural arch is completely flat in the anterior and central part of the vertebra, but a small, ridge-like protrusion can be found in between of the postzygapophyses. This protrusion likely represents a low neural spine, which gets progressively reduced along the caudal vertebral series of theropods.

Remarks: DfMMh/FV/105 is identified to be a theropod, and most likely a basal tetanuran, primarily on the basis of relatively elongate prezygapophyses (indicated by the position and shape of the postzygapophyses). In non-theropod dinosaurs, as well as more basal theropods such as Dilophosaurus wetherilli or Ceratosaurus sp., the prezygapophyses are generally much shorter in distal caudal vertebrae and do not extend far beyond the preceeding vertebra (Rauhut, 2003a).

Dinosauria Owen, 1842	
Theropoda Marsh, 1881	

Material: DfMMh/FV2/19, a small pedal phalanx III-1 (Figs. 2K–2P).

Description: DfMMh/FV2/19 is nearly completely preserved phalanx, with minor damage near the extensor groove, the right dorsal margin of the proximal facet, and around the ventral surface near the proximal end of the bone. The small phalanx is identified as a first pedal phalanx of the third digit of a theropod dinosaur, because of its broad and ventrally relatively flat proximal end, the non-saddle shaped proximal articulation surface, relatively long phalangeal shaft, and ginglymoid distal joint. It is unlikely that the phalanx represents a manual element, as the combination of a broad proximal end and a concave proximal joint surface are usually not found in manual phalanges (the first phalanx of the first manual digit usually has a saddle-shaped joint to mirror the condition of the first metacarpal, and the first phalanx of the manual second digit is usually relatively slender and less broad; e.g., Allosaurus fragilis: Gilmore, 1920; Madsen, 1976).

DfMMh/FV2/19 is relatively elongate and slender: it measures 10 mm proximodistally and a true phalangeal shaft separates the proximal and distal joints. This morphology is consistent with a first or second phalangeal position of the second or third digit, as other phalanges are usually stout and lack long phalangeal shafts (e.g., Allosaurus fragilis: Madsen, 1976). The phalangeal shaft of DfMMh/FV2/19 is transversely constricted in respect to the proximal and distal articulations, and is near circular in cross-section.

The proximal surface of DfMMh/FV2/19 is dorsally rounded and ventrally flat, therefore being ‘D’-shaped (Fig. 2M). It is five mm wide transversely, and three mm high dorsoventrally. The proximal articular surface is a single deep concavity. This is typical of first phalanges, which articulate with the broad trochlea of the metatarsalia, while more distally positioned phalanges usually have a saddle-shaped proximal joint that receives the condyles of preceding ginglymoid articulations typical for phalanges. The right dorsal margin of the proximal joint is unfortunately partly broken, but it seems that an extensor tubercle was very small if present at all. This again fits the morphological expectations for a first pedal phalanx.

The distal joint is ginglymoid, with a lateral and medial condyle (although it is not sure, which side is medial and which is lateral as it is currently not known whether the phalanx represents a left or right element) (Fig. 2P). Both condyles are subequal in size, and are separated from one another by a vertical intercondylar sulcus, that curves around the distal end of the bone. The condyles are slightly rotated anteriorly, so that they are ventrally stronger expanded than dorsally, and inclined outwards, so that the intercondylar sulcus gets broader anteroventrally. The sulcus opens into a shallow flexor groove posteriorly and ventrally. On the dorsal surface, just proximally to the condyles, there is a relatively deep extensor groove present, but its depth might be exaggerated by minor breakage around this part. Collateral ligament pits are hard to discern on the phalanx; on one side, it appears that no pit is present at all, and on the other side there is only a minor depression near the dorsal surface of the condyle (Fig. 2O).

Remarks: The phalanx DfMMh/FV2/19 is herein identified as belonging to an indeterminate theropod dinosaur. The relatively strongly ginglymoid distal articulation and ‘D’-shaped proximal articulation, combined with a relatively narrow and long phalangeal shaft are consistent with this interpretation. Phalanges of Jurassic turtles such as thalassochelydians have flatter shafts, often more broadly expanded proximal and distal ends, and the articular surfaces are less pronounced than in DfMMh/FV2/19 (e.g., Eurysternum wagleri, BSPG 1960 VIII 43). Pseudosuchian phalanges, for example from the Late Jurassic crocodyliform Alligatorellus sp., are usually more elongate and gracile than observed for DfMMh/FV2/19 (e.g., Tennant & Mannion, 2014).

Dinosauria Owen, 1842	
Theropoda Marsh, 1881	

Material: DfMMh/FV3/19, proximal part of a small right fibula (Figs. 4F–4J).

DfMMh/FV3/19 is a small fragment of a long bone, as it preserves parts of a shaft and one expanded terminal end. This specimen is herein identified as the proximal end of a right fibula of a theropod. The expanded proximal end of the specimen is relatively thick on one side, and thin-edged on the other side. The thicker side is interpreted to be the transversely expanded anterior side of the fibula (Fig. 4F), and the thin-edged side is interpreted to be the posterior margin of the fibula (Fig. 4I). However, fibulae are usually posteriorly stronger expanded than anteriorly, which is not the case in DfMMh/FV3/19. The presence of a large but shallow depression on what is interpreted as the medial side is consistent with the gross anatomy of a fibula.

The anterior and posterior sides of DfMMh/FV3/19 are slightly arched inwards towards the proximal end of the bone, so that the fibular head is gently crescentic. The surface of the articular facet is domed in its central part, and dips ventrally on the anterior side (Fig. 4H). The anterior margin of the articular facet forms a small lip that protrudes slightly anteriorly. The lateral surface of the fibular head is convexly rounded but becomes relatively flat towards the posterior side of the specimen. Just beneath the articular surface, the rounded anterior margin of DfMMh/FV3/19 is raised to a short tubercle or protuberance with slightly rugose surface texture that indicates the origin or insertion of some soft tissue structure. Posteriorly, the sharp-edged margin of the bone extends ventrally toward the shaft, and levels off reaching before the shaft.

On the medial side, the fibular head is characterized by a low, vaguely triangular concavity (Fig. 2J). The concavity is not well defined to either side, and spans more or less the entire space of the fibular head. The concavity narrows distally where it approaches the shaft, and finally vanishes just prior to a medial thickening of the fibular shaft.

The fibular shaft is broken shortly distal to the fibular head. The cross section of the break shows that the fibular shaft was circular in its proximal part.

Remarks: The described morphology of DfMMh/FV3/19 is consistent with its identification of an indeterminate theropodan fibula.

Discussion

Most German finds of Late Jurassic theropods are confined to the lagerstätten deposits of the Solnhofen area in Southern Germany, and include coelurosaurian theropods such as Juravenator starki (Gohlich & Chiappe, 2006), Archaeopteryx lithographica and closely related avian theropods (e.g., Foth, Tischlinger & Rauhut, 2014; Foth & Rauhut, 2017; Rauhut, Tischlinger & Foth, 2019), as well as the megalosauroid Sciurumimus albersdoerferi (Rauhut et al., 2012). Relatively complete theropod material from Northern Germany has been found in about 10 Myr older deposits from the Callovian, and belongs to the megalosauroid Wiehenvenator albati (Rauhut, Hübner & Lanser, 2016). On the basis of isolated teeth, Gerke & Wings (2016) found evidence for the presence of tyrannosauroids, allosauroids, megalosauroids, and ceratosaurs in the Langenberg Quarry. However, the findings of Van der Lubbe, Richter & Knötschke (2009), who reported on the presence of velociraptorine teeth from the Langenberg Quarry, could not be confirmed (Gerke & Wings, 2016). The fossils described in this contribution, interpreted as belonging to allosauroid, megalosauroid, ceratosaurian, and indeterminate theropods, represent the first body fossil evidence of theropods for the Langenberg Quarry. All of our material belongs to relatively small individuals. The ontogenetic stages of the material are currently unknown, but the presence of large theropod tracks in the Langenberg Quarry (Lallensack et al., 2015) demonstrates that large-bodied individuals were at least temporarily present in the habitat of today’s Langenberg Quarry. The fossil tooth, body fossil, and track record from Langenberg indicates a relatively high diversity of basal averostrans (i.e., ceratosaurs and basal tetanurans), which are rare elements of the Solnhofen archipelago limestone deposits. Despite the regional differences in faunal composition between different German basins, and although the Late Jurassic theropod fauna of Germany remains patchy, it is clear that all major groups of theropods that lived during the Late Jurassic were also present in Germany.

The Late Jurassic theropod dinosaur record in other parts of Europe is mostly similarly patchy, but also confirms the presence of several theropod lineages in Europe during the Late Jurassic. From the United Kingdom, diagnostic material is known from several formations and several stages of the Late Jurassic. For instance, the tyrannosauroid Juratyrant langhami is known from the Tithonian Kimmeridge Clay (Benson, 2008; Brusatte & Benson, 2013), whereas the allosauroid Metriacanthosaurus parkeri and the megalosauroid Eustreptospondylus oxoniensis are known from the older, Callovian–Oxfordian, Oxford Clay (Sadleir, Barrett & Powell, 2008; Carrano, Benson & Sampson, 2012).

Some of the best and most complete theropod material in Europe comes from Late Kimmeridgian–Tithonian formations in the Lusitanian Basin of Portugal, including the Lourinhã and Alcobaça formations. The Portuguese theropod fauna includes the allosauroid Allosaurus europaeus (Mateus, Walen & Telles Antunes, 2006), the ceratosaur Ceratosaurus sp. (Mateus & Antunes, 2000), the megalosauroid Torvosaurus gurneyi (Hendrickx & Mateus, 2014; Malafaia et al., 2017), the allosauroid Lourinhanosaurus antuneso (Mateus, 1998; Benson, 2010), the tyrannosauroid Aviatyrannis jurassica (Rauhut, 2003b). The faunal composition of the Portuguese record has been interpreted to be very similar to the much better documented equivalent North American fauna from the Morrison Formation (Mateus, 2006; Mateus & Antunes, 2000; Pol & Rauhut, 2012; Hendrickx & Mateus, 2014).

Conclusions

We present new occurrences of theropod dinosaurs from the Late Jurassic Langenberg Quarry of Northern Germany. The incomplete material can be assigned to certosaurian, megalosauroid, and allosauroid theropods. These identifications agree with previous reports of the presence of these theropod groups in the Late Jurassic of Northern Germany based on teeth. Although the Langenberg theropod fauna is not as rich as some other European localities, such as the Lourinhã Formation of Portugal, our findings confirm a varied dinosaur fauna in central Europe and add to our incomplete understanding of theropod diversity and evolution during the Late Jurassic of Europe.

We would like to thank Nils Knötschke and his team from the Dinosaurier-Park Münchehagen for collecting the material in the field and the exquisite preparation. Special thanks to the late Fabian von Pupka as well as Janna von Pupka, her team and family at the Rohstoffbetriebe Oker GmbH & Co. KG for the permission to access the Langenberg Quarry and for providing logistic support during fieldwork. We would like to thank Jonah Choiniere (University of the Witwatersrand), Oliver Rauhut (Bayerische Staatssammlung für Paläontologie und Geologie) and Mark Loewen (University of Utah) for providing images of comparative material. We also thank Elisabete Malafaia and Thomas Holtz for insightful reviews that improved an earlier version of this paper. Finally yet importantly, we would like to thank all excavation volunteers and preparators for their work on the Langenberg material.

Abbreviations

BSPG Bayerische Staatssammlung für Paläontologie und Geologie, Munich, Germany

BYU Brigham Young University, Provo, Utah, USA

DfmMh/FV Dinosaurier-Freilichtmuseum Münchehagen/Verein zur Förderung der niedersächsischen Paläontologie, Rehburg-Loccum, Germany

IVPP Institute of Vertebrate Paleomntology and Paleoanthropology, Beijing, China

JM Juramuseum Eichstätt, Eichstätt, Germany

MNN Museé National du Niger, Niamey, Niger

OUMNH Oxford University Museum of Natural History, Oxford, UK

SMNS Staatliches Museum für Naturkunde Stuttgart, Stuttgart, Germany

UC University of Chicago, Chicago, Illinois, USA

UMNH Utah Museum of Natural History, Salt Lake City, Utah, USA

Additional Information and Declarations

Competing Interests

Author Contributions

Data Availability

The authors declare there are no competing interests.

Serjoscha W. Evers performed the experiments, analyzed the data, prepared figures and/or tables, authored or reviewed drafts of the paper, and approved the final draft.

Oliver Wings conceived and designed the experiments, prepared figures and/or tables, authored or reviewed drafts of the paper, and approved the final draft.

The following information was supplied regarding data availability:

The materials described are accessioned and deposited in the DfmMh/FV (Dinosaurier-Freilichtmuseum Münchehagen/Verein zur Förderung der niedersächsischen Paläontologie, Rehburg-Loccum, Germany), where they can be examined by researchers: DfMMh/FV1/19, DfMMh/FV/343, DfMMh/FV/776, DfMMh/FV/287, DfMMh/FV/105, DfMMh/FV2/19, DfMMh/FV3/19.

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
