# Peer review of "Late Jurassic theropod dinosaur bones from the Langenberg Quarry (Lower Saxony, Germany) provide evidence for several theropod lineages in the central European archipelago"

_PeerJ, doi:10.7717/peerj.8437_

## Round 0.1 · original submission · Minor Revisions

Dear authors,

I have accepted the reviewers decision of ‘minor revisions’.

I look forward to receiving your revised manuscript.

·

Basic reporting

The authors provide important documentation of (admitted) sparse fossil specimens that nevertheless increase our knowledge of the terrestrial vertebrate fauna of the Late Jurassic of central Europe. I would say that none of their results are particularly surprising for a Late Jurassic dinosaurian assemblage (although the possible neovenatorid is intriguing, as this would be stratigraphic range extension of the clade), but such confirmations are significant nonetheless: they help to fill in the map of the biogeography of the ancient world.

It is a shame the material is not more complete, but of course the nature of the recovery (specimens found after blasting) would preclude actual complete skeletons. Their presence, however, might spur future systematic collection in this or comparable quarries in hopes of the discovery of more complete and intact fossils.

Experimental design

No comment

Validity of the findings

The identifications seem very reasonable: they are consistent with observed morphology. Of course, given how fragmentary they are and given the existence of homoplasy, it isn’t impossible (although unlikely) these all represent the same taxon that belongs to only one or none of these clades. If the elements assigned to Neovenatoridae are properly identified, this would seem to be the stratigraphically oldest members of this (otherwise Cretaceous) clade discovered this far.

Additional comments

In line 550 the authors use “about 10 Ma older” to specify “about 10 million years older.” While it is certainly true that some use Ma for durations, many authors prefer the scheme in which Ma refers to a date and Myr refers to a duration. Berggren and Van Couvering (1979,p. A506, footnote) specified that the abbreviation Ma refers to millions of years before present, and this was adopted by various stratigraphic codes. A discussion of this issue can be found in Aubry et al. (2009).

Aubrey, M.-P., J.A. Van Couvering, N. Christie-Blick, E. Landing, B.R. Pratt, D.E. Owen & I. Ferrusquía-Villafranca. 2009. Terminology of geological time: establishment of a community standard. Stratigraphy 6: 100-105.

Berggren, W.A. & J.A. Van Couvering. 1979. Quaternary. In Moore, R.C. (ed.) Treatise on Invertebrate Paleontology, Part A. Introduction. A505-A543. University of Kansas Press, Lawrence, KS.

·

Basic reporting

The authors describe several new and interesting occurrences of theropod dinosaurs from the Late Jurassic Langenberg Quarry of northern Germany, which is a poorly known record and thus this work adds important information for our knowledge of Late Jurassic continental ecosystem of central Europe.
The manuscript is clearly written in professional, unambiguous language, the literature references are updated, the context is well described.

Experimental design

This is an original research, the questions are well defined, relevant and meaningful, the investigation is rigorous and the methods are described with sufficient detail.

Validity of the findings

The figures are high quality, well labeled, and described, and the conclusions are relevant and well stated. However, I think that the identifications of some of the described elements need to be better supported (please see general comments below).

Additional comments

I made some notes directly on the manuscript. Apart from these notes, I have some additional comments and suggestions:

- It seems to me that the attribution of the ungual DfMMh/FV1/19 to a neovenatorid theropod is hard to justify. The ungual (and other) phalanges beside usually not well described in the literature show frequently similar features among different groups, which makes the identification of these elements extremely difficult. I agree that there are general similarities with some neovenatorid taxa, such as Neovenator and Australovenator, but it seems to me that there are also some differences, including the wider groove surrounding the articulation surface, which gives to the surface a more constricted appearance and the position of the flexor tubercle seems much more distal in DfMMh/FV1/19. Besides and despite some isolated elements tentatively related to this clade, the presence of neovenatorid theropods in the Late Jurassic of Lausaria needs to be confirm based on more complete specimens. For all these reasons, I would suggest a more open identification for this element.

- The same argumentation may be used for the phalanx DfMMh/FV/343. There are some features in this element that I think should be better described and analyzed. For example, the overall aspect of the phalanx, with a stout appearance in dorsal and ventral view, but relatively narrow in lateral view and the near flat proximoventral surface seem to me very different of the strongly offset proximal articular facet typical of most allosauroids. However, this may be also due to distortion. Could you please comment something about it? Other feature that I would like you to comment is the absence (or seems to me by the figure) of a dorsal groove adjacent to the distal articular facet. This groove is generally present in all pedal phalanges of most theropods even the most proximal ones.

- The identification of the chevron seems to me less problematic despite I agree with your caution maintaining the possible relationship with megalosauroids open. I just would like that you develop a little more the discussion on the presence of the medial keel on the anterior surface. It seems an interesting feature. A similar keel is present in some theropods, but as far as I know, it is usually much lower. Could you comment a little about the distribution of this feature among theropods?

- In the description of the caudal vertebra DfMMh/FV/105 could you provide a description of the ventral surface?

---

## Round 0.2 · accepted · Accept

Dear authors,

Based on your response to reviewers comments I have decided to ‘accept’ your manuscript.

Please note that it is the responsibility of the authors to ensure that the text makes sense linguistically, as PeerJ does not provide a full linguistic proofing check.

You will be contacted by production staff shortly about your proofs.

Thank you again for choosing PeerJ as your publication venue, and I hope you will use us again in the future.